# Evaluation of Joint Motion Sensing Efficiency According to the Implementation Method of SWCNT-Coated Fabric Motion Sensor

**DOI:** 10.3390/s20010284

**Published:** 2020-01-03

**Authors:** Hyun-Seung Cho, Jin-Hee Yang, Jeong-Hwan Lee, Joo-Hyeon Lee

**Affiliations:** 1BK21Plus Project, Clothing & Textiles, Yonsei University, Seoul 03722, Korea; hyunseung-cho@yonsei.ac.kr; 2Institute of Symbiotic Life-TECH, Yonsei University, Seoul 03722, Korea; sjinnie7@yonsei.ac.kr; 3College of Science and Technology, Konkuk University, Seoul 27478, Korea; jwlee95@kku.ac.kr; 4Deptartment of Clothing & Textiles, Yonsei University, Seoul 03722, Korea

**Keywords:** flexible fabric sensor, fabric strain gauge, single-walled carbon nanotube, joint motion sensing, garment structure, garment integrated sensing

## Abstract

The purpose of this study was to investigate the effects of the shape and attachment position of stretchable textile piezoresistive sensors coated with single-walled carbon nanotubes on their performance in measuring the joint movements of children. The requirements for fabric motion sensors suitable for children are also identified. The child subjects were instructed to wear integrated clothing with sensors of different shapes (rectangular and boat-shaped), attachment positions (at the knee and elbow joints or 4 cm below the joints). The change in voltage caused by the elongation and contraction of the fabric sensors was measured for the flexion-extension motions of the arms and legs at 60°/s (three measurements of 10 repetitions each for the 60° and 90° angles, for a total of 60 repetitions). Their reliability was verified by analyzing the agreement between the fabric motion sensors and attached acceleration sensors. The experimental results showed that the fabric motion sensor that can measure children’s arm and leg motions most effectively is the rectangular-shaped sensor attached 4 cm below the joint. In this study, we developed a textile piezoresistive sensor suitable for measuring the joint motion of children, and analyzed the shape and attachment position of the sensor on clothing suitable for motion sensing. We showed that it is possible to sense joint motions of the human body by using flexible fabric sensors integrated into clothing.

## 1. Introduction

A fabric sensor integrated into clothing senses the limb movements and is highly useful for sports monitoring applications, mainly on joint movements. Recent studies have focused on the development of motion sensors using fabrics and flexible materials, but studies on the efficiency of garment-integrated joint motion sensors have been relatively scarce. To monitor limb movements, the motion sensor needs to be integrated to the joint part of the clothing, and an elastic material is applied to the sensor to prevent interference with joint motion. When the fabric sensor is integrated into the garment, the arrangement and structural characteristics of the sensor are expected to affect motion-sensing performance. Therefore, the conditions under which the garment and sensor integration affect the efficiency of motion sensing need to be studied.

In this study, a stretchable fabric strain gauge sensor based on single-walled carbon nanotubes (SWCNTs) was developed and integrated into children’s clothing to evaluate the efficiency of motion-sensing performance. In our previous research [1], it was confirmed that the shape and attachment position of the sensor, in addition to the properties of the sensor material, are important factors for sensing efficiency when measuring the joint motion of the human body through fabric sensors integrated into clothes [2]. Thus, the effect of the shape and attachment position of the SWCNT-based stretchable fabric sensor on the joint motion-sensing performance is analyzed, and the fabric sensor structure and requirements to efficiently sense human joint movements just by wearing the garment were investigated.

### 1.1. The Sensor Material and Sensing Principle

Conventional strain gauges are resistive sensors, but they are used only for low elastic materials such as concrete or steel. The limbic movement monitoring clothing, meanwhile, requires highly elastic and flexible strain gauge. Therefore, in order to measure the intensity of the limbic movements via a garment, it is prerequisite to develop a resistive elongation sensors well-suited to textile application, accompanying with lightweight, flexible, elastic and washable characteristics.

For this research, we devised a fabric type piezoresistive sensor which is based on the conventional measurement principal as follows.
(1)$$R=\rho \times \frac{L}{S}$$
where *R* is the electrical resistance of the sample (Ω), ρ is the electrical resistivity of the material (Ω·m), *L* is the distance between measurement electrodes (m), and *S* is the section surface area of the sensor (m^2^) [3].

For the use of textile piezoresistive sensors, heavy and rigid materials such as metals are avoided in favor of light and flexible conductive materials. Two representative families of compounds, conductive polymer composites (CPCs) and intrinsically conductive polymers (ICPs), have an electro-mechanical behavior applicable to textile use. CPCs are based on a mix of a polymer matrix and electrical conductive filler. The fillers mainly used are divided into three groups, including carbon, organic conductive, and metal-based fillers. The most commonly used fillers for CPCs are carbon fillers, such as carbon black, carbon nanotubes, and graphite, as they are stable without oxidation under normal conditions, easy to use, and lightweight. SWCNT have emerged as a very promising new class of electronic materials. SWCNTs are nanometer-diameter cylinders consisting of a single graphene sheet wrapped up to form a tube. Since their discovery in the early 1990s, there has been intense activity exploring the electrical properties of these systems and their potential applications in electronics. Experiments and theory have shown that these tubes can be either metals or semiconductors, and their electrical properties can be compared to the best-known metals or semiconductors. Both metallic and semiconducting SWCNT possess electrical characteristics that compare favorably to the best electronic materials available. ICPs are inherently conducting or semi-conducting, owing to the presence of a conjugated π electron in their molecular structure. ICPs such as polypyrrole (PPy), polythiophene (PT), and polyaniline (PANI) present adequate combinations of electrical conductivity, stability, and processability.

In textile piezoresistive sensors that adopt the flexible conductive materials mentioned above, the sensing mechanism is based on at least one of the following three principles. The first sensing mechanism is based on the change in electrical resistance of the textile structure, which consists of a potentiometer per contact when stretched or deformed. If a very thin CPC or ICP layer coats some fibers of the textile sensor, its electro-mechanical behavior appears as a potentiometer per contact, rather than a sensor, where electrical resistance varies with dimension. The second sensing principle is associated to the change in electrical resistance of the sensor only affected by its geometrical change according to the elongation level. This case includes the textile elongation sensors made of ICPs, or CPC sensors, which are filled with a large amount of filler, much higher than the percolation concentration. The electro-mechanical behavior of a CPC-based sensor containing a large content of filler is mainly governed by variation of the sensor dimension, in elongation, represented in Equation (1). The third sensing mechanism is applied to the cases of electrical percolation type mechanical sensors (EPTMS) in CPC-based sensors, where elongation leads to the change in electrical resistance of the sensor, which is affected by filler disconnection in the sensor material. In this case, elongation induces a geometrical change owing to the aspect ratio (i.e., variation of *L* and/or *S*), and modification of the electrical resistivity of the sensor material as well. When the CPC filler content is near, but above, the percolation concentration, the electrical conductivity is highly sensitive to its volume change, owing to stretching. In this case, a small volume change of the sensor leads to numerous breakages in the conductive networks of the sensor material, causing a large variation in the overall resistivity of the CPC [3].

### 1.2. Research on Flexible Strain Sensors for Ambient Human Movement Monitoring

Wearable sensing through garment-integrated sensors has been promoted as an alternative, or a solution, coupled to traditional body sensing techniques. The most common wearable body movement sensing technique is the use of inertial sensing units (such as accelerometers and gyroscopes), which are often stiff, bulky, and possibly uncomfortable [4]. Bulky and uncomfortable wearable solutions for the wearer have been shown to affect the quality of the measured data [5] and may also affect the wearer’s attention and cognitive processes [6].

Shi et al. (2019) [7] researched the effect of treadmill on both gait and upper trunk movement characteristics by using two types of wearable sensors. Eight healthy male subjects are recruited to perform walking in two different conditions, 420-m straight overground walking (OW) and 5 min treadmill walking (TW), wearing the wearable sensors. Gait and upper trunk data were collected to comprehensively analyze the difference between two conditions. For the gait analysis, a set of inertial measurement unit was attached to each foot in the position close to toe by using elastic belts. Acceleration data acquired via the inertial sensors were to be utilized to analyze gait regularity and stability of swing phase. Insole sensors were put into subject’s shoes to acquire plantar pressure data for gait analysis during stance phase in walking. Linear and nonlinear analysis methods were used to evaluate the changes of spatiotemporal parameters, regularity and stability for both gait and upper trunk in walking. Paired *t*-tests are performed to compare linear and nonlinear features between TW and OW condition. Canonical correlation analysis was used to indicate the correlation between upper trunk movement characteristics and gait features in the aspects of spatiotemporal parameters and gait dynamic features. The research results showed that the treadmill could cause shorter stride length, less stride time and worsen long-range correlation of stride intervals, and therefore it could significantly increase the stability for both gait and upper trunk. Under the two walking conditions, the movement degree and regularity of upper trunk appeared to be very similar. It means that the treadmill does not change the kinematic characteristics of upper trunk. Canonical correlation analysis results showed that treadmill could reduce the correlation between gait and upper trunk features. The researchers interpreted these results that people tended to walk more cautiously to prevent the risk of falling and neglected the coordination between gait and upper trunk when walking on the treadmill.

Yamada et al. (2011) [8] developed a wearable and stretchable strain sensors fabricated from thin films of aligned single-walled carbon nanotubes (height, 1 mm; thickness, 6 mm; length, 16 mm). The SWCNT films were laid side by side, with a 1 mm overlap, arranged perpendicular to the strain axis on a flat and elastomeric dog-bone-shaped backing structure made of PDMS (poly(dimethylsiloxane), PDMS; thickness, 1 mm), with depositing the support electrodes of Ti(3 nm)/Au(100 nm)/Ti(10 nm) at both ends of the substrate for strain sensor characterization. Each film was wet with a droplet of isopropyl alcohol, which flattened the film (thickness, 400 nm) to the substrate; this allowed the SWCNTs to be packed into a highly densely packed solid form (density, 0.46 g cm^−3^; occupancy, 42%; SWCNT spacing, 4.1 nm). This process resulted in the development of a strong van der Waals contact with the substrate, achieved without any additional mechanical pressure. The adhesion strength was measured as ~12 N^−2^ and was sufficient to bear large strain. Representative resistivity–strain data resulting in a monotonic increase up to 280% strain (strain speed, 1 mm min^−1^), at which point the PDMS substrate ruptured, demonstrated the potential use of this device as a gauge to measure strains 50 times more than conventional metal strain gauges. The gauge factors of the strain sensor was derived to be 0.82 (0 to 40% strain) and 0.06 (60 to 200%); in comparison, conventional metal gauges have a factor of 2.0 (5% maximum strain) and thermal plastic elastomer with 50 wt% carbon black a factor of 20 (80% maximum strain). The carbon-nanotube strain sensor exhibited superior durability and stability. At 100 and 150% strain and a strain speed of 6 mm s^−1^, the strain sensor electrical response remained nearly unchanged after 10,000 cycles, while the sensor was stable for 3300 cycles at 200% strain until the substrate ruptured, showing that the performance was limited by the substrate. To demonstrate the potential of the SWCNT films in wearable devices, the researchers assembled the carbonnanotube sensors on stockings, bandages and gloves to fabricate devices that could detect different types of human motion, including movement, typing, breathing and speech.

Gioberto & Dunne (2014) [9] introduced the characterization of a novel garment-integrated stitched sensor response to bends and fabric folds with different morphology (the kind of unconstrained folding that is seen in garments during body movement) and explored the influence of the characteristics of the fabric substrate on the sensor response. The repeatability, accuracy, and relations observed in controlled scenarios under different conditions show the ability of the sensor to detect bending effectively, while preserving wearer comfort, garment aesthetics, and ease of production. They tested a garment-integrated stitched sensor for five types of folds, stitched on five different weights of un-stretchable denim fabric and analyzed the effects of fold complexity and fabric stiffness, under both un-insulated and insulated conditions. The results showed that insulation improves the linearity and repeatability of the sensor response, particularly for higher fold complexity. Stiffer fabrics showed greater sensitivity, but less linearity. The sensor response amplitude was larger for more complex fold geometries. The utility of a linear bending response (insulated) and a binary shorting response (un-insulated) was discussed. Overall, the sensor exhibited excellent repeatability and accuracy, particularly for a fiber-based, textile-integrated sensor [10,11,12]

Cochrane et al. (2007) [13] developed a CPC-based textile sensor, able to measure their strain deformations. The sensor developed in this paper is based on a thermoplastic elastomer/carbon black nanoparticle composite integrated into a textile substrate. The researchers focused on importance of the integration of a sensor on a textile substrate which would not modify its general behavior. The material used for the sensor was a composite based on a thermoplastic elastomer (Evoprene 007 (EVO), a Styrene-Butadiene-Styrene (SBS) co-polymer) for the polymer matrix, and carbon black (CB) nanoparticle for the conductive filler, which presented general mechanical properties strongly compatible with those of the textile substrate of a thin lightweight Nylon fabric. Two techniques for CPC processing were firstly investigated: the conventional melt-mixing process and the solvent-mixing process, which was found to be better adapted for this particular application. The optimization of the process in terms of filler concentration relevant to the percolation theory was investigated. A dramatic decrease in resistivity was observed for the same given conductive filler content in both cases at 7.3 vol.-% of filler contents. This critical concentration was inferred to correspond to the percolation volume in the percolation theory explaining that the electric charges would form electro-conductive channels at such point and a transition of the material from electrically insulating to conductive would occur. In terms of conductivity, the solvent-processed blends seemed to be better than the melt-processed ones near the percolation threshold, probably because of a better dispersion of the particles in the solvent process due to the lower viscosity of the solution stemming from their primary form of liquid or gel. Considering that the resistivity of the system should be in a measurable range (<100 Ω·m), the optimal CPC blend should contain at least 27 vol.-% of CB particles. The conductive filler concentration used was determined to be 27.6 vol.-% by compromise between sensor sensitivity and resistivity value. In the second step, the researchers developed a textile strain gauge by applying the solvent-mixing process and performed an electromechanical characterization to demonstrate the adaptability and correct functioning of the sensor as a strain gauge. The obtained electrical resistivity vs. strain curve was divided in two regions: the first one corresponds to a strain below 15%, where the sensor response was not linear. The non-linearity of the behaviour was explained by three factors including the geometrical influence of the sensor, the change in the percolated system structure, and the re-arrangement of the system’s network. In the second region, for strain values greater than 15%, the sensor response was practically linear. The gauge factor K for this linear zone was be determined to be K to be 80. This exceptionally high value of K for the CPC sensor in comparison with the classical metal gauges was explained by two factors, the change of geometry of the sensor and a change in the percolation network of the system. Finally, the influence of environmental factors, such as temperature and atmospheric humidity, on the sensor performance was investigated. The results show that the sensor’s electrical resistance is particularly affected by humidity. This behavior was discussed in terms of the sensitivity of the carbon black filler particles to the presence of water.

Mattmann et al. (2008) [14] presented a textile strain sensor able to measure a large strain of at least 80%. For the development of the stain sensitive conductive fiber a mixture of a thermoplastic elastomer (TPE) and carbon black particles was used. It consisted of a mixture of 50 wt% thermoplastic elastomer (TPE) and 50 wt% carbon black particles (1.21 g cm^2^, 32 vol-%) was used to produce the textile strain sensors with a diameter of 0.315 mm, resulting in a resistance of approximately 700 Ω/cm. at this filling level. The thread-shaped sensors with the same characteristics were attached to two different textiles (486 Meryl (88% PA, 12% lycra, knitted) and Keller AG 88018 (49% PA, 51% EL, woven). The Meryl knit was about three times more elastic than the woven textile from Keller AG. The sensor properties were examined in terms of relaxation behavior, hysteresis, working range, dependency on strain rate, long term cycling, ageing, and washability. The sensor was cycled between 0% and 80% strain at a speed of 200 mm/min and waiting times at minimal and maximal strain of 2 min. When the strain was kept constant, it relaxed by 1.5 kΩ while the total range was 17 kΩ, which resulted in an inaccuracy of 8.8% caused by the relaxation behavior. For the period between 0% and than 10% strain level, the electrical resistance didn’t follow the applied strain, staying at the resistance level which corresponded to a strain of about 10%. This was explained to be caused by a temporary deformation of the textile due to the large strain applied. The sensor has a high sensitivity of 1.25 kΩ /mm (=250 Ω%/strain) and a gauge factor of ∼20 at a sensor length of 2 cm. Therefore, it was recommended to use the sensor in the working range (above 10% to 80% strain). It was also found that pre-stretching of the textile sensor was necessary to ensures stable sensor properties. For working range, the sensor showed a linear resistance response to strain, a small hysteresis (the maximal hysteresis error of ±3.5% (7%) in strain at 16 kΩ), no ageing effects, and a small dependence on the strain velocity. It was reported that washing of sensors several times in a conventional washing machine did not influence the sensor properties. This study also showed an example application, where 21 strain sensors were integrated into the back region of a tight-fitting clothing. With this garment, 27 upper body postures performed by eight participants could be recognized with an accuracy of 97%.

Shyr et al. (2014) [15] fabricated a strain-resistance sensor by using elastic conductive webbing consisting of carbon coated fibers and elastic fibers. PAC fibers (Polyamide fiber coated with carbon particles) having a diameter of 50 μm was used as the conductive fiber. Fifteen PAC fibers were twisted with a polyester yarn at a rate of 80 twists per meter to form a conductive yarn. A Lycra fiber was cross-wrapped over two polyester yarns to form an elastic yarn. The elastic conductive webbing had a plain structure, 8 mm wide by 2 mm thick. The warp of the webbing was made up of 32 conductive yarns and five elastic yarns, and the weft was made of one strand of the conductive yarn. The developed strain sensor showed to have high resistance sensitivity, low tensile hysteresis, as well as high linearity and repeatability of the relationship between strain and resistance without resistance hysteresis. By applying this sensor, they developed a wearable gesture-sensing device, which was designed for monitoring the flexion angle of the elbow and knee movements. The resistance of the elastic conductive webbing to the flexion angle had a good linear relationship (coefficient of determination (*R*^2^) of the linear regression curve was 0.98) during the stretch-recovery cycles within 30% strain. The relationship between flexion angle and resistance of the wearable gesture sensing device was calibrated and established using the gesture sensing apparatus with a variable resistor and a protractor which were worn on the same positions and synchronously recorded during elbow and knee movements. The flexion angle-resistance equations of the wearable gesture sensing device were then established as: *y* = −37*x* + 595 (*R*^2^ = 0.96) for elbow movement, and *y* = −19*x* + 280 (*R*^2^ = 0.97) for knee movement. In comparison with the results of the gesture sensing apparatus with a variable resistor and a protractor, the results obtained from their wearable gesture sensing device were found to be consistent with those from the gesture sensing apparatus. This result indicated that the wearable gesture sensing device based on a textile strain sensor successfully monitored the flexion angles during elbow and knee movements.

Gibbs & Asada (2005) [16] developed a wearable joint monitoring sensor capable of continuous, day-to-day monitoring. A novel technique for incorporating conductive fibers into flexible, skin-tight fabrics surrounding a joint was developed. The purpose of this study was to develop a wearable joint sensor capable of continuous monitoring by measuring single or multi-axis joint angles in a reliable and non-intrusive way. The researchers devised the sensor by attaching the conductive fiber strands to a nonconductive form-fitting (elastic) fabric substrate. To avoid the erroneous measurements cased by misalignment of a sensor from use to use, and recalibration problem in every wearing, the researchers decided to use an array of multiple threads in a known pattern per motion-axis, which could be facilitated in determining a sensor’s offset from calibration by a template-matching algorithm. The sensor array could be taken off and put back on an individual for multiple uses, with the sensors automatically calibrating themselves each time. Resistance changes across these conductive fibers were measured and directly related to specific single or multi-axis joint angles after an initial, one-time calibration. The two specific predictor models (the linear and quadratic models) were adopted for the calibration of a set of sensor in this study. After preliminary experiments for lower body monitoring, it was derived that this sensing device could be effective for monitoring joint motion of the hip and knee. The developed pants type wearable sensors were comfortable, and acceptable for long-term wear in everyday settings. The single axis knee joint experiment resulted in that the pants type sensors were able to quite accurately capture the knee joint movement patterns in all types of motion including higher frequency motion. It represented the average rms error between the pants type sensor and the potentiometer using the linear predictor was 5.4°, while 3.2° using the quadratic predictor. The double axis experiment consisting of a sequence of semi-random hip joint movement showed that the pants type sensing device was able to capture the hip joint movement patterns over time. The average rms error between the pants type sensors of hip flexion angle and that of the goniometer was 2.5° using the linear predictor, and 2.4° using the quadratic predictor. For hip abduction, there errors were 2.1° using the linear predictor, and 1.7° using the quadratic predictor. The overall research results indicated the feasibility of this pants type sensor, with higher accuracy measurements of joint motion for both a single-axis knee joint and a double axis hip joint when compared to the standard goniometer used to measure joint angles.

Seyedin et al. (2015) [17] presented a scaled-up fiber wet-spinning production of electrically conductive and highly stretchable PU/PEDOT:PSS fibers for the first time. The PU/PEDOT:PSS fibers possess mechanical properties appropriate for knitting various textile structures as well as conductivity. Based on their investigation on knit textiles, the researchers developed a knitted textile strain sensor that exhibited low resistance, high sensitivity, high stability, and a large sensing range. A highly stable sensor response was observed when four PU/PEDOT:PSS fibers were co-knitted with a polyurethane yarn. For this, the PU/PEDOT:PSS containing 13.0 wt % PEDOT:PSS loading with an electrical conductivity of ∼9.4 S cm^−1^ and a Young’s modulus of ∼23.5 MPa, tensile strength of ∼22.7 MPa, elongation at break of ∼345%, and toughness of ∼39.8 MJ m^−3^ was used in this work. The measurement result showed that the absolute value of the gauge factor increased with the number (ply) of PU/PEDOT:PSS fibers in the knitted textile from ∼−0.2 for single-ply, to ∼−0.5 for double-ply, and ∼−1.0 for four-ply, which indicated that the strain sensing properties of the developed knitted textiles were dependent upon the numbers (plys) of PU/PEDOT:PSS fibers used in knitting. The resistance of the knitted textile sensors appeared to be highly stable after 500 cycles demonstrating reproducible strain sensing properties during the cyclic tests. The strain sensing behavior of the textile structure comprising of four-ply PU/PEDOT:PSS fibers and a Spandex yarn was evaluated at applied strains of up to 180%. The electromechanical behavior of the knitted textiles tested under different levels of applied strain can be categorized into three distinct zones (0–80% (zone 1), 80–160 % (zone 2), and>160% (zone 3)). Above 160 % strain in zone 3, individual fibers were also stretched and contribute to strain sensing. The breakage of the conducting filler network within the individual PU/PEDOT:PSS fibers was related to the sudden increase in resistance after cyclic stretching at 180%. However, at 160% no gaps could be observed between the fibers in the textiles and the fibers were relatively parallel to the stretching direction suggesting that individual fibers were also stretched. When the applied strain was below 160%, the textile extension was only through the elongation of loops and legs, as well as bending of the heads. The results presented here suggest that the knitted textile sensor can be used for applications that require strain sensing up to 160%. This sensing range was found to be significantly higher than the previous reports on coated textiles (10–80%) and knitted silver plated yarns (40%). The knitted textile responded well to the magnitude of bending deformations, demonstrating potential for remote strain sensing applications. The feasibility of an all-polymeric knitted textile wearable strain sensor was demonstrated in applications such as personal training and rehabilitation following injury

In other study, force sensitive resistors (a polymer thick film) and fabric stretch sensors (a conductive carbon-loaded rubber) have been used to provide alternative methods to detect muscle activity. Amft et al. (2006) [18] developed these sensors to detect the contractions of arm muscles during four types of hand and arm activities including upward movement of lower arm, outward bending of hand, opening and closing of hand, and grasping of heavy object with right arm. They established two setups for the motion detection, attaching the two force sensitive resistors to the lower arm onto the belly centres of it, and wrapping the fabric stretch sensors around the lower arm covering the belly centres. The measurements resulted in that the stretch sensor signal changed when the action was executed for any of the four actions and that muscle activity monitoring was more clear and feasible in using FSR.

## 2. Research Methods

### 2.1. Development of Joint Motion Sensor

#### 2.1.1. Implementation of Conductive Fabric

In this study, SWCNT, one of the CPC materials, was selected as the main material for the fabric motion sensor, and a flexible fabric-type joint motion sensor based on this material was implemented by the following method (Figure 1).

To implement a conductive fabric for joint motion, a conductive and stretchable slurry was made to coat on the base fabric layer. This conductive slurry is composed of single-walled carbon nano black powder (specific surface area 100~200 m^2^/g), water, urethane emulsion, and other materials. All of these three materials were used to prepare a single-walled carbon nano black slurry. To prepare the single-walled carbon nano black slurry, pure water with 100% weight ratio and carbon nano black powder with 60% weight ratio were measured, dispersed in a tank, and stirred. A urethane emulsion was added to the dispersed slurry at a weight ratio of 25% and further dispersed once more into the continuous type bead mill. A single-walled carbon nano black slurry was added to the knitted fabric (77% polyester/23% spandex) of a base fabric. The fabric was put in a desiccator in a vacuum state, so that the single-walled carbon nano black slurry sufficiently deposited and coated it. The deposited fabric was placed in a dry curing machine (max. 200 °C) and cured at 120 °C for 1 h to complete the preparation for a stretchable and conductive fabric as the material of the fabric motion sensor. The resistance per unit area of the final fabric motion sensor developed through the series of processes was measured. As a result of 10 random measurements of the whole sample, the average resistance per unit area (10 × 10 cm) was between 280 Ω and 290 Ω, with a standard deviation of 2.5 Ω. Figure 2 shows images of the fabric motion sensor implemented through the above process, taken by a scanning electron microscope S-4800 (HITACHI) at a magnification of 200 to 1,000,000.

#### 2.1.2. Joint Motion Sensor

A sensor that measures the limb joint motion of the human body using the SWCNT-based conductive and stretchable fabric was developed in this study. That is, a knitted fabric of SWCNT membrane-coated polyester and urethane blend was laser-cut and integrated into clothing.

The sensors were fabricated in two shapes, rectangular and boat-shaped and each sensor was integrated into children’s clothing of the same fabric and structure (Figure 3). The rectangular sensor was fabricated with a length and width of 40 mm, 5 mm. On the other hand, the boat-shaped sensor had a length, sensor center width, and both end widths of 40 mm, 10 mm, and 5 mm. The boat-shaped sensor was used in this study in addition to the typical rectangular strain gauge sensor for the following reason. When the limb joint motion is measured through the sensor attached to the garment, the center portion of the fabric sensor is excessively stretched owing to the protrusion of the joint at the time of the motion, while both ends of the sensor may be less stretched. This can cause that the motion signal measured from the sensor generates a lot of noise (TEXTILE MOTION SENSOR, A METHOD OF FABRICATING THE SAME, AND CLOTHING SYSTEM INCLUDING THE SAME / Korean Patent registration number: 10-18956940000, Date of patent registration; 30/08/2018). Therefore, in this study, by deforming the shape of the conventional rectangular strain gauge sensor into a boat type with a wider center, it is possible to attenuate the phenomenon that the center part of the textile motion sensor is overstretched owing to the joint protrusion at joint motion. In other words, we tried to find a way to improve the accuracy of joint motion sensing.

As an experiment for evaluating the sensitivity of the sensor, the elongation and relaxation of the fabric sensor sample fabricated in the rectangular form (100 × 20 (mm)) was repeated 10 times. The results are shown in Figure 4.

### 2.2. Test Clothing

The fabric sensors were integrated into the clothing by welding method. Four types of test garment were fabricated by welding the fabric sensors to children’s clothing using different combinations of the two shapes and two positions of the sensor, which were the two research variables in this study.

When sensing joint motion through the sensor attached to the clothing, the force applied to the fabric sensor during motion changes depending on its position in the clothing. In addition, if the position of the sensor in the clothing moves owing to joint motion, then the accuracy of the motion signal measured by the fabric sensor is reduced. The stability of the position of the fabric sensor is largely affected by the fit of the clothing and the position of the sensor on the clothing. Therefore, in this study, the level of fit of the clothing integrated with the fabric sensor was controlled to be at a ‘snugly-fitted level’, that is, the daily activity level, and the attachment position of the fabric motion sensor was manipulated in order to investigate the difference in joint motion sensing depending on two different positions. The two sensor attachment positions were the ‘joint position’, in which the center of the fabric sensor was located at the most protruding position of the joint; and the ‘position 4 cm below the joint’, in which the center of the fabric sensor was separated from the most protruding position of the joint by placing it 4 cm below the joint (Figure 5).

### 2.3. Method and Procedure

#### 2.3.1. Subjects

Three boys, average age of six years (Table 1), were selected as subjects in the experiment study. Prior to the experiment, a consent form was signed by the guardian of each child to comply with research ethics regulations. The fitting evaluation was performed on each test clothing by examining the sensor shape and attachment position while the child was wearing the test clothing. The subjects were given a thorough explanation regarding the experimental protocol for measuring the motion and were supervised to practice it repeatedly.

#### 2.3.2. Test Procedure

After putting the motion-sensing test clothing developed in this study on the children as test subjects, flexion-extension motions of the arms and legs were repeated and changes in the resistance of the fabric motion sensor due to elongation and contraction were measured. The experimental variables were the two fabric sensor shapes (rectangular or boat-shaped), and the attachment positions of the sensor on the clothing (at the elbow and knee joints or at 4 cm below the elbow and knee joints). For all subjects, the test was performed at two angles of motion, 60° and 90°, at a rate of 60°/s. The flexion-extension motions were repeated 10 times for both angles, which was considered as one set. The voltage output from the motion sensor integrated into the clothing during three sets of repetitive motions was used as experimental data. In addition, a commercial low-speed acceleration sensor (low-g accelerometer) was attached near the fabric sensor, which simultaneously measured and generated reference data to verify the reliability of the fabric motion sensor during arm and leg movements.

To prevent the children’s individual motion abilities and habits from intervening, a multi-purpose muscle strength measuring device (Con-Trex MJ, CMV AG Co., Switzerland) was used to control the joint motion speed and joint motion range (Figure 6). The measurement circuit was set at 5 V as the reference voltage and at 10 kΩ for fixed resistance. The data output through the fabric sensor at the flexion-extension motion was sampled at intervals of 20 ms via the measuring instrument of the Data Logger (GT342). The experiment setup and signal processing flow was shown in Figure 7.

## 3. Results and Discussion

The morphological characteristics of the peak-to-peak voltage (Vp-p) signal waveform were examined to analyze the sensing efficiency of the fabric motion sensor depending on the shape and attachment position. The evaluation criteria for sensing performance included the baseline value (Vmin) uniformity of the Vp-p, amplitude of the Vp-p signal, and uniformity of the Vp-p signal. The hysteresis of the sensor was analyzed by changes in the baseline value (Vmin) of Vp-p. The more uniform the baseline, the better the recovery of the fabric sensor. The hysteresis decreased during joint motion, showing high sensing reproducibility and stability. The clarity of the signal was also analyzed using the average value of the Vp-p signal amplitude during joint movement. The reproducibility and accuracy of the sensor were also verified by calculating the average deviation of Vp-p; the smaller the average deviation of the Vp-p signal amplitude, the more uniform the detection signal and the higher the reproducibility and accuracy of the sensor. Furthermore, to evaluate the reliability of joint movement sensing through the fabric sensor, the consentaneity of the output signals from the acceleration sensor and fabric sensor during limb movement were compared.

### 3.1. Arm Movement

In this study, the sensing efficiency depending on the shape of the fabric sensor and its attachment position in clothing was evaluated with the above-mentioned criteria through the output voltage from the flexion-extension operation of the arm. Figure 8, Figure 9, Figure 10, Figure 11, Figure 12 and Figure 13 show graphical representations of the changes in the voltage measurement output and baseline value (Vmin) of the 60° and 90° flexion-extension movements of the arm joint of child subject A.

The morphological characteristics of the motion signal graphs in Figure 8, Figure 9, Figure 10 and Figure 11 show a larger signal at the 90° angle of motion than that at 60°. It was observed that the fabric sensor has the fundamental characteristics of a general strain sensor, which outputs a larger signal when further stretched or elongated. Measurements of the elbow flexion and extension motion of the arm using this sensor show that the overall signal was uniform and stable, and the consentaneity with the acceleration sensor was as high as 95.8% (95.6% for the 60° flexion-extension motion, and 95.9% the for 90° flexion-extension motion). However, as shown in Figure 8, when the sensor was attached to the joint position with 90° angle of motion, a minute double-peak signal was observed.

As shown in Table 2 and Figure 12 and Figure 13, the baseline changes during the 60° and 90° flexion-extension movements of the arm indicate the smallest baseline deviation at the joint position of the boat-shaped sensor during 60° arm motion, and at 4 cm below the joint position of the rectangular sensor during the 90° arm motion, thus demonstrating stability. This reveals that the restoration force of the fabric sensor was satisfactory and has small hysteresis, which is a requirement for the sensor to obtain the most reliable signal.

Meanwhile, in order to quantitatively analyze the effects of shape and attachment position of the fabric sensor on sensing performance, the average value and average deviation of the difference between Vmax and Vmin were calculated. First, to verify the clarity of the measured signal against the signal magnitude, the average value of the difference between Vmax and Vmin of the motion measurement test data was calculated and was used as a variable to perform an independent sample *t*-test using SPSS Win 21.0.

As presented in Table 3 and Table 4, the boat-shaped sensor has significantly larger motion signals than the rectangular sensor. The motion signal obtained from the sensor attached at the joint was significantly larger than that placed 4 cm below the joint. With regard to the clarity of the sensed motion signal (amplitude of Vp-p signal), out of the four combinations of sensor shapes and attachment positions, the size of the joint motion sensing signal became the largest and the signal was most clearly identified when the boat-shaped sensor was attached to the joint position.

Next, the mean deviation of the difference in Vp-p was calculated for the output of the arm motion measurements at 60° and 90°, and the waveform characteristics were obtained. The mean deviation value was used as the variable for an independent sample *t*-test to verify the statistical difference. In other words, the uniformity of the measured signals was verified by analyzing the difference in Vp-p was how much far apart from the average value. Because the movement speed and angle were controlled constantly by the Con-Trex MJ equipment, small average deviations of flexion-extension motion in all the experiments in this study means that the fabric sensor signal was uniform and the reproducibility of the sensing signal was high.

As presented in Table 5 and Table 6, there are significant differences in motion sensing performance depending on the shape of the sensor and attachment position. A more uniform signal was generated when the rectangular sensor was attached 4 cm below the joint, and higher signal reproducibility was obtained. This result shows that the measured motion signal (Vp-p signal amplitude) is smaller when the rectangular sensor was attached 4 cm below the joint, but the sensing result is more reproducible with higher sensing efficiency.

### 3.2. Leg Movement

Figure 14, Figure 15, Figure 16, Figure 17, Figure 18 and Figure 19 show graphical representations of the changes in measured voltage output and baseline values (Vmin) during 60° and 90° flexion-extension movements of the knee joint of child subject A.

The morphological characteristics of the sensing output graphs of leg motion in Figure 14, Figure 15, Figure 16 and Figure 17 show larger Vp-p than arm motion; however, a double-peak signal appeared at 90° motion angle in all sensor test conditions, regardless of the sensor shape and attachment position. The double-peak signal seems to be related to the sensitivity of the sensor because the fabric sensor was relatively more stretched as the motion angle increased. The area of the sensor attached to the knee area was larger than that of the elbow, and the degree of elongation during 90° flexion-extension of the legs was larger than that at 60°.

In other words, the double-peak signal appeared when the fabric sensor was over-stretched. If the sensor stretches beyond its length during 60° flexion-extension leg movement, it could exceed the threshold; therefore, the attachment position at the knee may need to be adjusted for the sensor to become less elongated. In general, however, uniform and stable signals were observed, and 97% of the measured signals were in consentaneity with the motion signals of the accelerometers (97.1% and 97% during 60° and 90° flexion-extension motions, respectively).

Table 7 and Figure 18 and Figure 19 show the baseline changes during 60° and 90° flexion-extension motion of the leg. The signals obtained showed the smallest deviations during both the 60° and 90° leg motions when the rectangular sensor was attached to the joint; thus, this is the ideal configuration to obtain a signal with low hysteresis and high reliability.

Table 8 and Table 9 present the results of statistical analysis of the clarity of the measured signals through the Vp-p signal magnitude. The signal amplitude was larger in the boat-shaped sensor than in the rectangular sensor, and when the sensor was attached 4 cm below the joint than when attached to the joint. In other words, the boat-shaped sensor attached 4 cm below the joint showed the clearest signal during leg motion.

The boat-shaped sensor showed higher efficiency during both arm and leg movements in terms of the signal clarity through the amplitude analysis. The center area of this sensor was twice as wide as that of the rectangular sensor; hence, the Vp-p signal amplitude is expected to increase because the elongated area also increased. Meanwhile, it was found that that the signal amplitude according to attachment position differs for each arm and leg movement.

The average deviation of the difference in Vp-p was calculated for the output data of the leg motion measurement at 60° and 90° as in the arm motion, and the statistical significance of the difference in sensing efficiency of different sensor shapes and positions was also analyzed. The results are presented in Table 10 and Table 11. The difference in motion sensing efficiency owing to sensor shape was statistically significant, but no significant difference was found in relation to the sensor attachment position. The sensor shape was considered as more reproducible, as the rectangular sensor showed more uniform output value than the boat-shaped sensor. The sensor attachment position on the clothing was not statistically significant, but the sensor attached 4 cm below the joint had a slightly smaller value than the average, indicating a more uniform signal value.

## 4. Conclusions

In this study, a SWCNT-based fabric motion sensor was studied to develop a wearable sensor capable of measuring the joint motion of the human body with higher efficiency. The test clothing were manipulated into two sensor shapes (rectangular and boat-shaped) and two attachment positions (at the elbow and knee joints, and 4 cm below the joints). The analysis of the output values (Vp-p) of joint movement tests on children showed that the fabric sensor developed in this study has high reproducibility and stability. Furthermore, the fabric sensor is suitable for sensing children’s joint movements by verifying the agreement between the motion sensed by the fabric and acceleration sensors. The sensing efficiency depending on sensor shapes and attachment positions on clothing was also quantitatively analyzed, and the following results were obtained.

First, the sensor indicated high signal reproducibility and reliability when the rectangular sensor was attached 4 cm below the joint during arm movement. When the rectangular sensor was attached regardless of the attachment position during the leg movement, which results in more uniform sensing signal. The sensor attachment position had no effect on the uniformity of the signal during leg movement. Overall, in order to obtain highly reproducible and stable motion signals during arm and leg movements, the most important requirement is to attach the rectangular sensor 4 cm below the joint.

Second, with regard to the amplitude of the sensed motion signal and its clarity, the boat-shaped sensor yielded more efficient results than the rectangular sensor during both arm and leg movements, mostly because the center area of the boat-shaped sensor was almost twice that of the rectangular sensor, and the relatively stretched area during joint movement was also expected to increase.

Therefore, in order to improve the sensing efficiency while obtaining a clear signal, it is necessary to analyze the performance of the sensor by closely examining the bone and muscle characteristics of the human body as well as further manipulating the area, length, and attachment position of the sensor. In this study, we developed a sensor suitable for measuring children’s limb joint movements and analyzed the conditions of sensor shapes and attachment positions for efficient motion sensing. The study is meaningful in that it showed that it is possible to sense movement in separate body parts through a flexible fabric sensor integrated into the clothing.

## Figures and Tables

**Figure 1 sensors-20-00284-f001:**
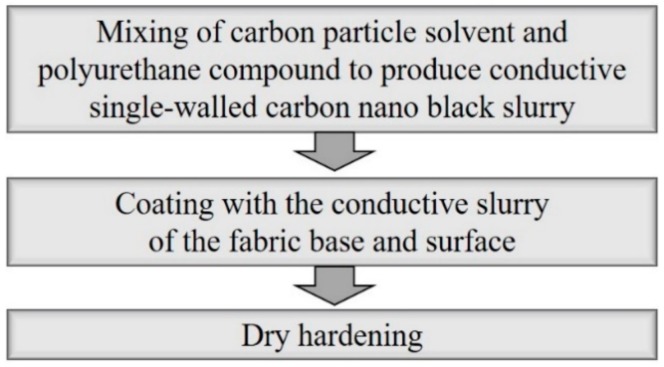
Implementation of conductive fabric.

**Figure 2 sensors-20-00284-f002:**
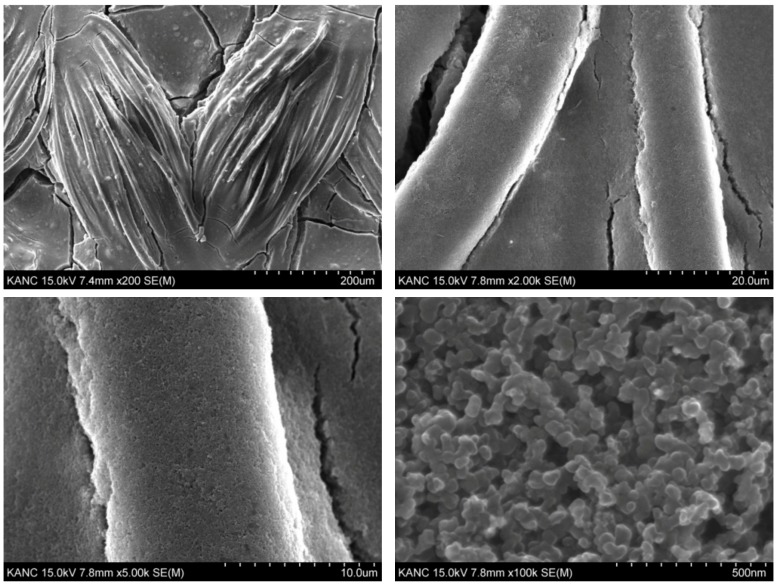
SEM image of the SWCNT-coated sensor materials.

**Figure 3 sensors-20-00284-f003:**
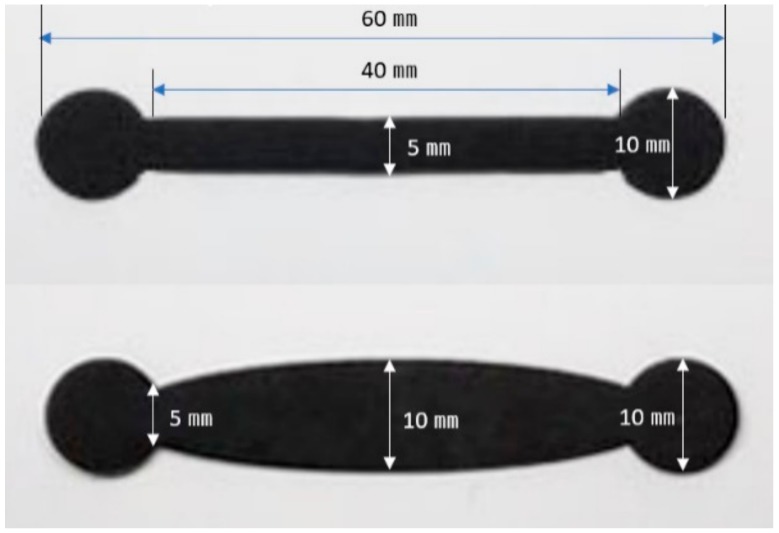
Two-types of fabric sensors (up: rectangular sensor, down: boat-shaped sensor).

**Figure 4 sensors-20-00284-f004:**
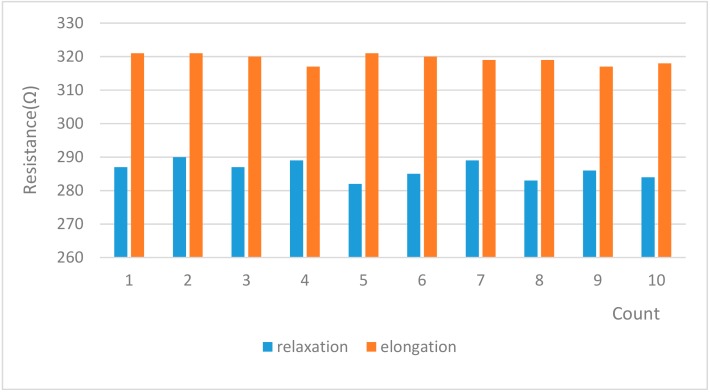
The resistivity change versus displacement.

**Figure 5 sensors-20-00284-f005:**
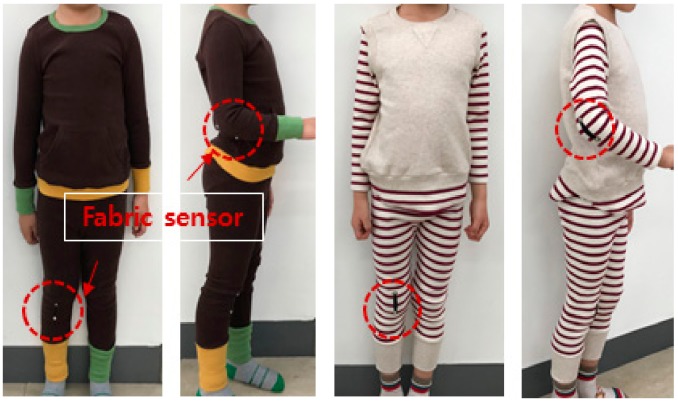
Examples of developed test clothing with integrated fabric motion sensor.

**Figure 6 sensors-20-00284-f006:**
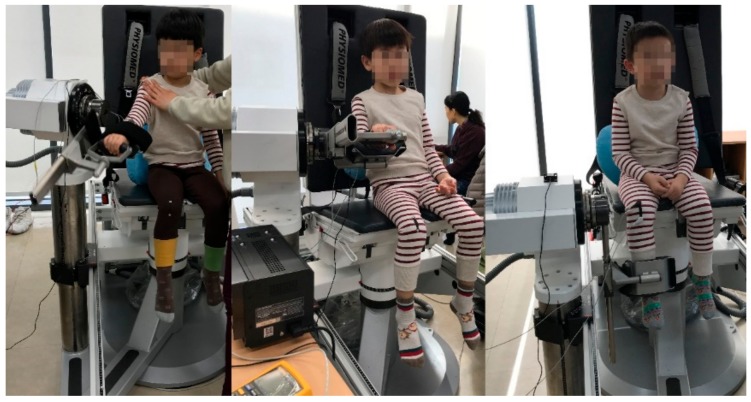
Flexion-extension motion measurement of arms and legs.

**Figure 7 sensors-20-00284-f007:**
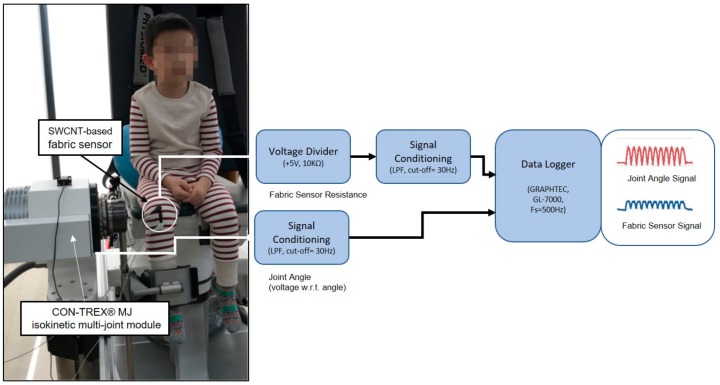
Experiment setup and signal processing flow.

**Figure 8 sensors-20-00284-f008:**
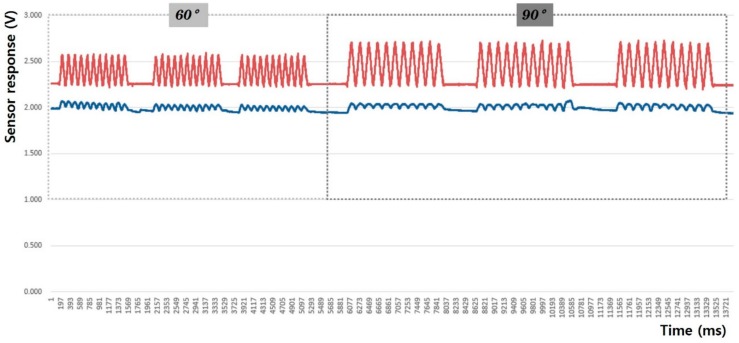
Arm motion signal from the rectangular sensor at the joint (up: joint angle signal, down: fabric sensor signal).

**Figure 9 sensors-20-00284-f009:**
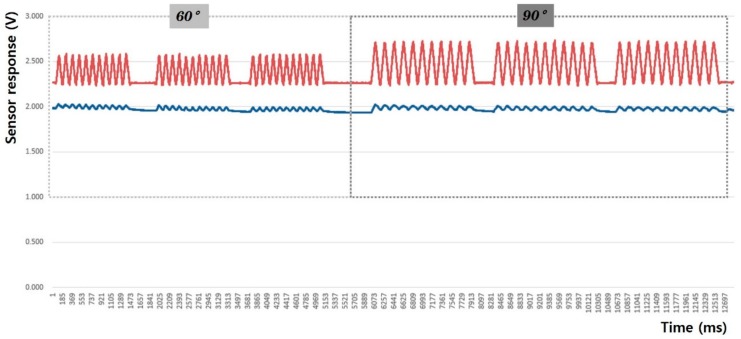
Arm motion signal from the rectangular sensor attached 4 cm below joint (up: joint angle signal, down: fabric sensor signal).

**Figure 10 sensors-20-00284-f010:**
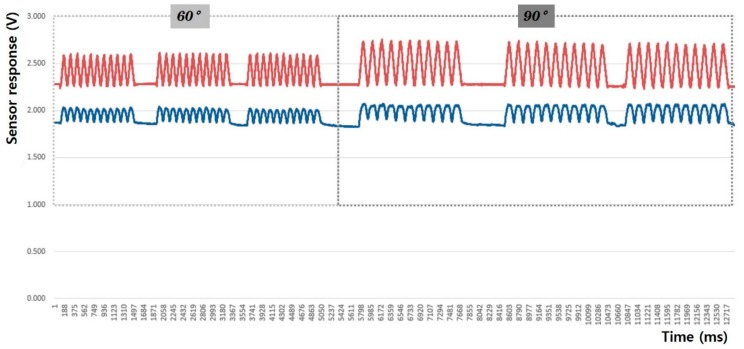
Arm motion signal from the boat-shaped sensor at the joint (up: joint angle signal, down: fabric sensor signal).

**Figure 11 sensors-20-00284-f011:**
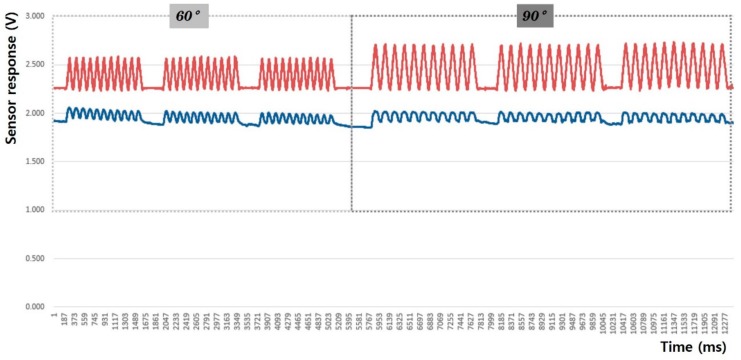
Arm motion signal from the boat-shaped sensor attached 4 cm below the joint (up: joint angle signal, down: fabric sensor signal).

**Figure 12 sensors-20-00284-f012:**
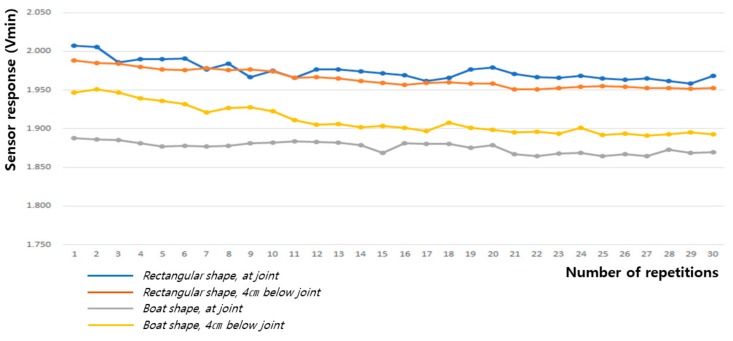
Change in baseline value (Vmin) during 60° flexion-extension motion of the arm joint.

**Figure 13 sensors-20-00284-f013:**
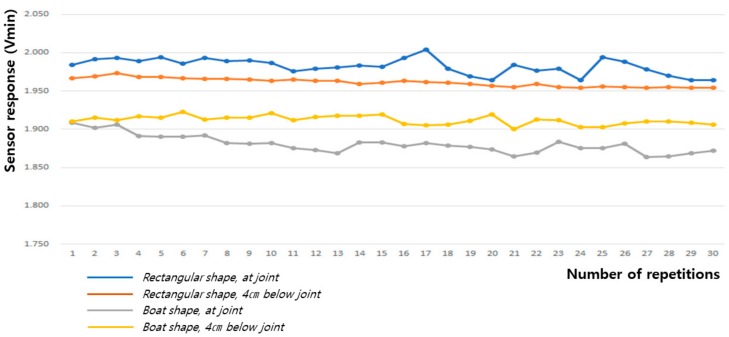
Change in baseline value (Vmin) during 90° flexion-extension motion of the arm joint.

**Figure 14 sensors-20-00284-f014:**
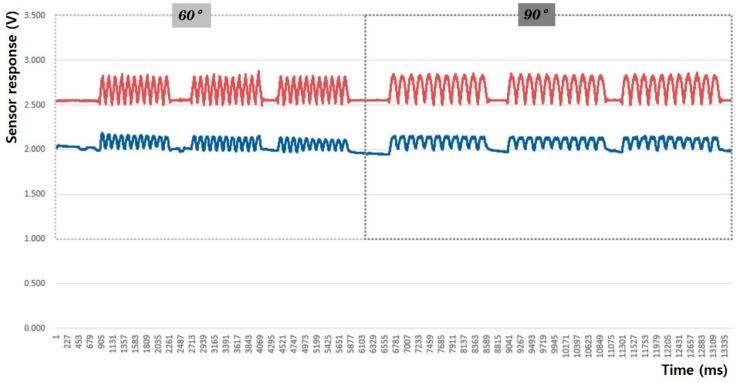
Leg motion signal from the rectangular sensor at the joint (up: joint angle signal, down: fabric sensor signal).

**Figure 15 sensors-20-00284-f015:**
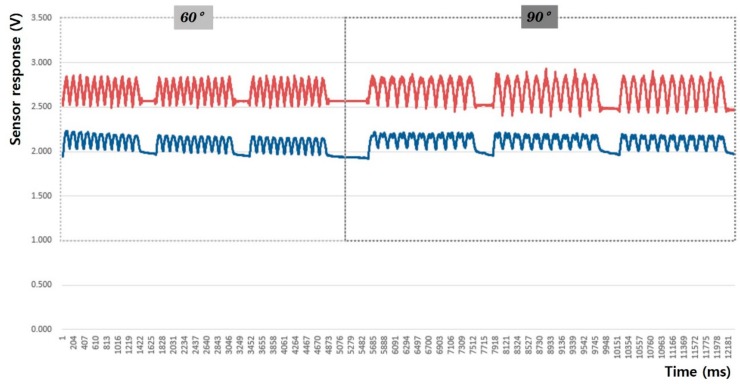
Leg motion signal from the rectangular sensor attached 4 cm below the joint (up: joint angle signal, down: fabric sensor signal).

**Figure 16 sensors-20-00284-f016:**
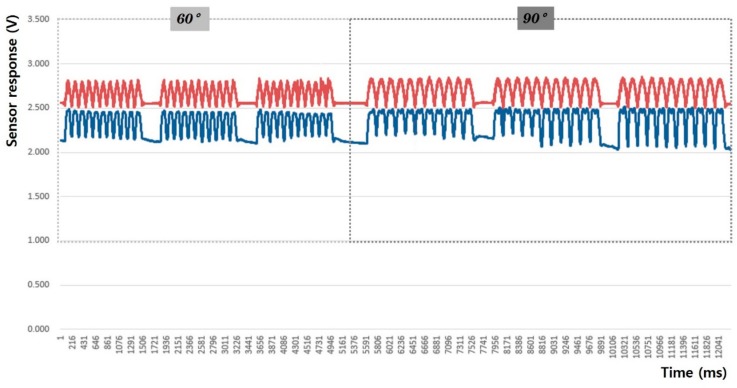
Leg motion signal from the boat-shaped sensor at the joint (up: joint angle signal, down: fabric sensor signal).

**Figure 17 sensors-20-00284-f017:**
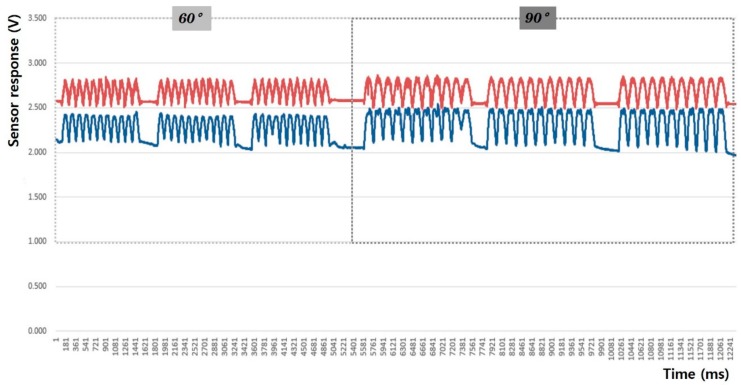
Leg motion signal from the boat-shaped sensor attached 4 cm below the joint (up: joint angle signal, down: fabric sensor signal).

**Figure 18 sensors-20-00284-f018:**
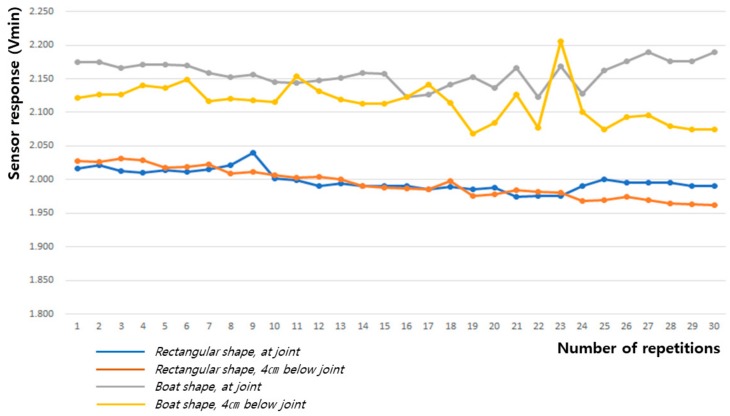
Change in baseline value (Vmin) during 60° flexion-extension motion of the knee joint.

**Figure 19 sensors-20-00284-f019:**
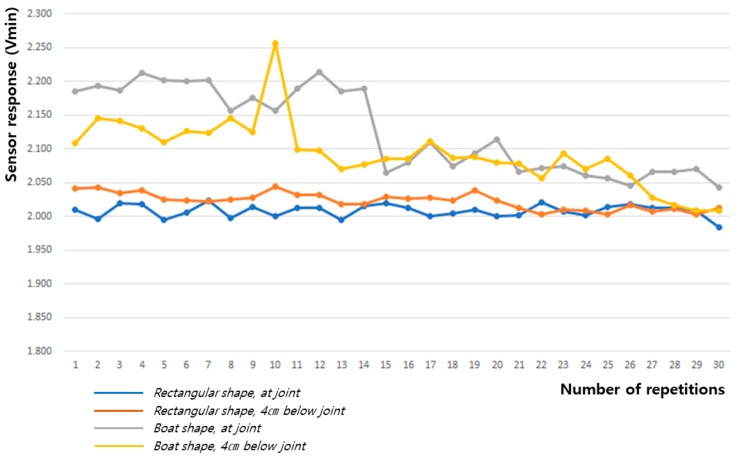
Change in baseline value (Vmin) during 90° flexion-extension motion of the knee joint.

**Table 1 sensors-20-00284-t001:** Body types of the subjects (somatotype).

	Subject A	Subject B	Subject C
Height (cm)	116.4	116.1	122.5
Weight (kg)	23.6	17.3	22.4
Elbow circumference (cm)	18.5	17	18
Knee circumference (cm)	27	24	27
**BMI**	17.4	12.8	14.9

**Table 2 sensors-20-00284-t002:** The baseline value (Vmin) for the sensing signal during the flexion-extension motion of the arm joint.

Movement Angle	60° Flexion-Extension Motion				90° Flexion-Extension Motion			
Textile Sensor	Rectangular Shape, at Joint	Rectangular Shape, 4 cm below Joint	Boat Shape, at Joint	Boat Shape, 4 cm below Joint	Rectangular Shape, at Joint	Rectangular Shape, 4 cm below Joint	Boat Shape, at Joint	Boat Shape, 4 cm below Joint
Mean	1.975	1.965	1.876	1.912	1.983	1.961	1.881	1.912
Standard deviation	0.013	0.012	0.007	0.019	0.010	0.005	0.012	0.006

**Table 3 sensors-20-00284-t003:** Clarity of the arm motion sensing signal according to the sensor shapes.

	Sensor Shape	*N*	*Mean*	*SD*
Sensing performance	rectangular sensor	360	0.04404	0.014306
	boat-shaped sensor	360	0.11895	0.035578
*t(p) −37.067(0.000) ****				

* *p* < 0.1, ** *p* < 0.05, *** *p* < 0.01.

**Table 4 sensors-20-00284-t004:** Clarity of the arm motion sensing signal at different sensor attachment positions.

	Attached Position	*N*	*Mean*	*SD*
Sensing performance	at joint	360	0.09032	0.048895
	4 cm below joint	360	0.07266	0.041681
*t(p) 5.215(0.000) ****				

* *p* < 0.1, ** *p* < 0.05, *** *p* < 0.01.

**Table 5 sensors-20-00284-t005:** Difference in sensing efficiencies of the arm motion according to sensor shapes.

	Sensor Shape	*N*	*Mean*	*SD*
Sensing performance	rectangular sensor	360	0.00691	0.05756
	boat-shaped sensor	360	0.01608	0.012791
*t(p) −12.396(0.000) ****				

* *p* < 0.1, ** *p* < 0.05, *** *p* < 0.01.

**Table 6 sensors-20-00284-t006:** Difference in sensing efficiencies of the arm motion for different sensor attachment positions.

	Attached Position	*N*	*Mean*	*SD*
Sensing performance	at joint	360	0.01450	0.012495
	4 cm below joint	360	0.00849	0.08038
*t(p) 7.673(0.000) ****				

* *p* < 0.1, ** *p* < 0.05, *** *p* < 0.01.

**Table 7 sensors-20-00284-t007:** The baseline value (Vmin) for the sensing signal during the flexion-extension motion of the knee joint.

Movement Angle	60° Flexion-Extension Motion				90° Flexion-Extension Motion			
Textile Sensor	Rectangular Shape, at Joint	Rectangular Shape, 4 cm below Joint	Boat Shape, at Joint	Boat Shape, 4 cm below Joint	Rectangular Shape, at Joint	Rectangular Shape, 4 cm below Joint	Boat Shape, at Joint	Boat Shape, 4 cm below Joint
Mean	1.999	1.995	2.157	2.116	2.009	2.023	2.130	2.096
Standard deviation	0.015	0.368	0.018	0.029	0.008	0.012	0.061	0.047

**Table 8 sensors-20-00284-t008:** Clarity of the leg motion sensing signal according to the sensor shapes.

	Sensor Shape	*N*	*Mean*	*SD*
Sensing performance	rectangular sensor	360	0.14298	0.039962
	boat-shaped sensor	360	0.26747	0.073849
*t(p) −28.131(0.000) ****				

* *p* < 0.1, ** *p* < 0.05, *** *p* < 0.01.

**Table 9 sensors-20-00284-t009:** Clarity of the leg motion sensing signal difference sensor attachment positions.

	Attached Position	*N*	*Mean*	*SD*
Sensing performance	at joint	360	0.18603	0.089584
	4 cm below joint	360	0.22442	0.077828
*t(p) −6.137(0.000) ****				

* *p* < 0.1, ** *p* < 0.05, *** *p* < 0.01.

**Table 10 sensors-20-00284-t010:** Difference in sensing efficiencies of the leg motion according to sensor shapes.

	Sensor Shape	*N*	*Mean*	*SD*
Sensing performance	rectangular sensor	360	0.01136	0.009866
	boat-shaped sensor	360	0.03259	0.025616
*t(p) −14.674(0.000) ****				

* *p* < 0.1, ** *p* < 0.05, *** *p* < 0.01.

**Table 11 sensors-20-00284-t011:** Difference in sensing efficiencies of the leg motion for different sensor attachment positions.

	Attached Position	*N*	*Mean*	*SD*
Sensing performance	at joint	360	0.02269	0.021783
	4 cm below joint	360	0.02126	0.022450
*t(p) 0.868(0.386)*				

* *p* < 0.1, ** *p* < 0.05, *** *p* < 0.01.

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
