# Peer review of "Evaluation of Joint Motion Sensing Efficiency According to the Implementation Method of SWCNT-Coated Fabric Motion Sensor"

_sensors, 2020, doi:10.3390/s20010284_

Round 1

Reviewer 1 Report

It is very interesting to use fabric motion sensors to measure human arm and leg motions. However, the author needs to rewrite the introduction part and analyze the related works deeply. The experimental results are not analyzed in details. Please refer to "The Effect of Treadmill Walking on Gait and Upper Trunk through Linear and Nonlinear Analysis Methods" to add figures to explain the results. And using statistical methods to analyze the results. In addition, it is better for the author to give an entire framework for the paper. More details about the fabric motion sensors and also the subjects should be given.

Author Response

We truly appreciate the comments from reviewer intending to improve the quality of this article. We tried to correct everything according to the reviewer's instructions. In addition to the content modifications, figure corrects and additions, all in red letters. If you have any additional improvements or would like to make any other modifications, please give me some instruction. Thank you for your careful consideration.

Reviewer 2 Report

Evaluation of joint motion sensing efficiency according to the implementation method of SWCNT-coated fabric motion sensor

This study reports a development of a textile piezoresistive sensor suitable for measuring the joint motion of children, and an analysis of the shape and attachment position of the sensor on clothing suitable for motion sensing. Authors showed that it is possible to sense joint motions of the human body by using flexible fabric sensors integrated into clothing.

The authors gave an important state of the art. However, it is difficult to understand or to find what is the added value of the reported study?Is it the using of the composite or the proposed geometries? Are these sensors more sensitive than others proposed in this sate of the art?

The 2 figures 1 and 2 showed the SEM images of the material composed the sensor and the fabricated sensors. I would suggest to add the scheme and the principle working of the 2 proposed geometries with dimensions.

The figure 5 shows the real experimental conditions. However, I would suggest to add an electrical and details of the characterization proposed system for the measurements (inputs and outputs signals, methodology). A scheme of the test bench will give more technical clarification to this study.

In addition, the responses of the 2 proposed sensors geometries are not given. Indeed, I would suggest to add the resistivity change versus displacement. This is very important to estimate the sensor characteristics such as sensitivity in order to compare the performances of the two geometries: (rectangular and boat-shaped) and also to the proposed existing sensors in the state of the art.

Author Response

(The authors gave the same response as above.)

Round 2

Reviewer 1 Report

The author has changed the manuscript according to the comments. It can be accepted. The author needs to revise it according to the format of the journal. 

Author Response

Thank you very much for accepting the revised paper.
I will do my best to study harder.

Reviewer 2 Report

I would suggest to specify in figure 4 (Figure 4. The resistivity change versus displacement) if the measured resistivity change is based rectangular sensor or boat-shaped sensor? Because the 2 geometries should have different relaxation and elongation?

The authors have addressed all my concerns (my last questions) in this revised version. Thank you.

Author Response

Figure 4 shows the results from a 10x2 cm rectangular sensor sample.

We included it in that part of the article.

Thank you very much for your careful attention.